# Plasmatic Profile of Pregnancy-Associated Glycoprotein (PAG) during Gestation and Postpartum in Sarda and Lacaune Sheep Determined with Two Radioimmunoassay Systems

**DOI:** 10.3390/ani10091502

**Published:** 2020-08-25

**Authors:** Martina De Carolis, Olimpia Barbato, Gabriele Acuti, Massimo Trabalza-Marinucci, Noelita Melo de Sousa, Claudio Canali, Livia Moscati

**Affiliations:** 1Department of Veterinary Medicine, University of Perugia, 06126 Perugia, Italy; martina.decarolis.87@gmail.com (M.D.C.); gabriele.acuti@unipg.it (G.A.); massimo.trabalzamarinucci@unipg.it (M.T.-M.); claudio.canali@unipg.it (C.C.); 2Laboratory of Animal Endocrinology and Reproduction, Faculty of Veterinary Medicine, University of Liege, 4000 Liege, Belgium; noelitamelo@gmail.com; 3Istituto Zooprofilattico Sperimentale dell’Umbria e delle Marche, via G. Salvemini 1, 06126 Perugia, Italy; l.moscati@izsum.it

**Keywords:** RIA, PAG, gestation, sheep, postpartum, breed, multiple pregnancy, single pregnancy

## Abstract

**Simple Summary:**

Nowadays the need to optimize and maximize the productivity of dairy sheep leads farmers to plan lambing in specific periods in order to avoid economic loss due to nonproductive animals. The goal is to diagnose pregnancy at early gestation in order to minimize the costs of unproductive animals and to properly formulate rations for the energy requirements of gestating or lactating animals at the same time. Moreover, early pregnancy diagnosis, as well as the possibility of distinguishing between single and multiple gestations, enables farmers to plan the management of lambing well in advance thus minimizing manpower requirements. This study showed, for the first time, the plasmatic profile of PAG (pregnancy-associated glycoproteins) in Sarda and Lacaune ewes during gestation and the postpartum period using two different radioimmune assay (RIA) systems, thus enhancing our knowledge regarding PAG concentrations in domestic ruminants. Moreover, it showed that for both breeds these RIA systems are capable of distinguishing pregnant from nonpregnant ewes starting from day 18 of gestation. Furthermore, the rapid disappearance of PAG concentration following lambing did not require the use of a cut-off limit in postpartum animals as a means for detecting a new pregnancy.

**Abstract:**

This study was carried out to determine ovine pregnancy-associated glycoprotein (oPAG) levels in the plasma of Sarda and Lacaune ewes throughout gestation and in the first month postpartum, using two heterologous radioimmunoassays (RIA-706 and RIA-srPool) and to study the correlations between PAG levels and fetal gender and number. On Day 18 of pregnancy, PAG concentrations were detected in 90.1% and 80.8% of Sarda pregnant ewes with RIA-706 and RIA-srPool, respectively; and in 90% and in 75% of Lacaune pregnant ewes with RIA-706 and RIA-srPool, respectively. From Day 30, PAG concentrations were detected in all pregnant ewes by using both RIA methods. In the postpartum period, the PAG concentrations in Sarda ewes decreased rapidly reaching minimal levels (<1 ng/mL) on day 28 using both RIA-706 and RIA-srPool. In Lacaune ewes, PAG-706 levels were higher than PAG-srPool from parturition until the last day of observation (Day 28 postpartum). It was also observed that mean concentrations were higher in multiple than in single pregnancies in Sarda and Lacaune ewes. Moreover, due to the rapid disappearance of PAG concentration following lambing, a cut-off limit in postpartum animals was not required as a means for detecting a new pregnancy.

## 1. Introduction

Nowadays the need to optimize and maximize the productivity of dairy sheep leads farmers to plan lambing in specific periods in order to avoid economic loss due to nonproductive animals. The goal is to diagnose pregnancy at early gestation in order to minimize the fodder wasted on unproductive animals and to properly formulate rations for the energy requirements of gestating or lactating animals at the same time. Moreover, early pregnancy diagnosis, as well as the possibility of distinguishing between single and multiple gestations, enables farmers to plan the management of lambing well in advance thus minimizing manpower requirements.

In this regard, over the past 30 years, techniques such as radioimmunoassay (RIA) for pregnancy detection have been developed using pregnancy-associated glycoproteins (PAGs) which are glycoproteins belonging to the aspartic proteinase family and are synthesized in the ruminant trophectoderm [1]. Glycoproteins associated with pregnancy in sheep are represented by PAGs (OPAG) [2,3,4] and SBU-3 [5]. In ovine placenta, 11 cDNA coding for distinct PAGs (ovPAG-1 to ovPAG-11) were identified at different gestational periods, thus confirming the multiplicity and temporal expression of PAG molecules in ruminant placenta [1,6,7,8,9,10].

These molecules are present in maternal plasma of sheep in concentrations detectable with RIA methods as early as 18–20 days after conception, [3,11,12,13,14,15,16] and by the enzyme immune assay (EIA) method [17,18,19], as well as in other species of ruminants [1,20,21,22]. Throughout pregnancy, PAG concentrations in these species differ according to the breed, fetal number, sex and birth weight [3,23,24,25,26,27,28,29], as in other ruminants [30].

To our knowledge, no studies have yet been carried out on PAG plasma concentration profiles during gestation and the postpartum period in Sarda and Lacaune ewes, two amongst the most important dairy sheep present in Europe.

Sarda is the main sheep breed raised in Italy and it has been selected over the years for milk production [31,32]. The breeding system is typically characterized by one lambing per year and the mating season starts in early summer (or early autumn for ewe lambs). However, especially for yearlings, total fertility rate is not higher than 75%. When lambing occurs in early spring, since the breeding system is based on pasture, milk production is positively affected by high forage availability. However, strategies to favor out-of-season lambing are encouraged to ensure cheese availability for the market over the year [33,34].

The Lacaune breed originates from the Roquefort area, Southern France. It is the main French dairy sheep breed and has been very efficiently selected during the last 40 years [35]. The Lacaune is now a high milk yield sheep that, contrary to what happens for the Sarda breed, is also appreciated for lamb growth rate and the characteristics of its meat. Moreover, out-of-season lambing is common, and the breeding system is not necessarily based on pasture, being characterized by a large use of conserved forages and concentrates.

Both breeds would benefit from an early pregnancy diagnosis to reduce economic losses and increase productivity [36].

The aim of this study was to investigate the concentrations of plasma PAG using two heterologous radioimmunoassays (RIA-706 and RIA-srPool) in Lacaune and Sarda sheep throughout gestation and after parturition, and to study the correlations between PAG levels and fetal gender and number. Correlations between concentrations measured with the aforementioned RIA systems, their ability to detect PAG molecules during pregnancy and their half-life were also investigated.

## 2. Materials and Methods

### 2.1. Animals and Samples

Thirty-five Sarda ewes weighing 42 ± 1.4 kg (mean ± SEM) at mating, and thirty-five Lacaune ewes weighing 46 ± 1.2 kg (mean ± SEM) at mating from a single flock were enrolled in this study. All of the ewes were in their first lactation during the period September–January of 2008. The ewes were housed and managed at the Azienda Zootecnica Didattica of the University of Perugia. The experimental site is approximately 15 km southwest of Perugia, Italy (latitude: 41°34′ N and longitude: 14°39′ E) at an elevation of approximately 650 m above sea level. The area has a Mediterranean climate with an annual rainfall of approximately 650 mm, distributed mainly during late autumn and winter, and mean maximum and minimum temperatures ranging between 15.7 and 8.4 °C over the last 40 years.

Blood samples were withdrawn from the jugular vein and placed into EDTA-coated tubes (Sarstedt^®^, Numbrecht, Germany). The samples were collected on Days 0 (day of mating), 18, 30, 45, 60, 90, 120 after mating and during the postpartum period, starting within 12 h of birth (*p*) and on Days 7, 14, 21, 28. Plasma was obtained by centrifugation (2500× *g* for 15 min) immediately after collection and was stored at −20 °C until assay.

### 2.2. Experimental Design

The two groups of ewes were housed in two separate 15 × 5 m straw-bedded pens. The animals were subjected to the same management practices and had feed and water readily available in feeders and troughs in order to ensure an adequate intake of food and water. The basal diet was composed of alfalfa hay (2.0–2.5 kg/head/day, depending on the breed) which was supplemented with pelleted concentrate (crude protein: 180 g/kg; neutral detergent fiber: 149 g/kg as fed) during lactation and the last two months of pregnancy (800 g and 600 g/head/day, respectively) (Mignini&Petrini Spa., Petrignano di Assisi, Perugia, Italy) to meet nutrient requirements according to Cannas (2004) [37].

The ewes were synchronized with intravaginal sponges containing 40 mg fluorogestone acetate (Cronogest sponge, Intervet, Milan, Italy) for 12 days. On sponge withdrawal, the ewes were injected (i.m.) with 350 UI PMSG (Pregnant mare serum gonadotropin) ((Folligon, Intervet, Milan, Italy) and 36 h later two rams of proven fertility were introduced (September 2008) to each group for one day and the females were then separated after mating.

Conception was assumed to have occurred 149 days before parturition, which is the average gestation period for sheep. The number and sex of lambs were recorded.

The lambs were kept with the mothers until the end of the trial.

The animals in this trial were supervised in compliance with Italian laws (DL 27 January 1992, n°116) and regulations regarding experimental animals. The experimental design was performed according to good veterinary practices under farm conditions.

### 2.3. Progesterone Assay

The radioimmunoassay analyses (progesterone, PAG) were performed at the University of Perugia (Department of Veterinary Medicine). Samples from Sarda and Lacaune ewes were assayed for progesterone using an extraction step described elsewhere [38,39]. Progesterone was extracted from plasma with diethyl ether and the efficiency of the extraction procedure was monitored by addition of a tracer amount of (3H) progesterone. The efficiency of the ether extraction ranged from 85 to 95%. Extraction was carried out with 0.2 mL plasma and each sample was assayed in duplicate. Volumes of 0.8 mL of distilled water and 3 mL of diethyl ether were then added to each sample and centrifuged at 1000× *g* for 10 min. Following freezing, the supernatant was discarded and 1 mL of borate buffer with 10% ethanol was added to all samples. Standard curve dilution was prepared using plain tubes for total counts and nonspecific binding. A 0.1 mL volume of increasing concentrations of calibrators (P4), 0.1, 0.25, 0.5, 2, 5, 10, 20 and 40 ng/mL was added. Reference samples (0.5 ng/mL and 10 ng/mL) were also added as quality controls. ^3^H-labeled progesterone (0.1 mL) and 0.1 mL of progesterone antibody were added to the experimental plasma samples extracted. Incubation was carried out at 4 °C for at least 4 h. The radioactivity antibody bound P4 from free P4 was separated by centrifugation following dextran-charcoal adsorption. The tubes were transferred to a beta-counter (Tri-carb 2100 TR, Packard) to be counted. The minimum detection limit (MDL) was 0.09 ng/mL. Intra and interassay coefficients of variation (CV) were 8 and 12%, respectively.

### 2.4. Pregnancy-Associated Glycoprotein Assays

Two different radioimmunoassay systems (RIA-706 and RIA srPool) obtained from the methods previously described in detail by Perenyi et al. [40,41] and Barbato et al. [12] were used to measure pregnancy-associated glycoprotein concentrations. All the assays were performed in Tris buffer containing 1% BSA (Fraction V, ICN Biochemicals Inc., Aurora, OH, USA). Measurements were performed in polystyrene tubes and all of the incubations were performed at room temperature (20 to 22 °C). Bovine PAG 67 kDa preparation (boPAG_67kDa_, accession number Q29432) was used as standard and tracer for all assays [30,42].

The ewe samples (0.1 mL, in duplicate) were assayed in a preincubated system. In short, 0.1 mL of each sample, or appropriate standard dilution, were aliquoted into duplicate assay tubes and diluted with 0.1 mL and 0.2 mL of Tris-BSA buffer, respectively. In order to minimize nonspecific interference of plasma proteins, 0.1 mL of bovine PAG-free plasma was added to all standard tubes. Subsequently the antisera (1:80,000 for RIA-706 and 1:50,000 for RIA-srPool) were added and the tubes were incubated overnight at room temperature before adding radio-labelled PAG [40]. Iodination (Na-I^125^, Amersham Pharmacia Biotech, Uppsala, Sweden) was carried out according to the Chloramine T method [43].

Samples with higher PAG concentrations than the estimated standard dose for which the percentage B/B0 was 20% (ED20) were reassayed in nonpreincubated systems in which the standard curves ranged from 0.8 to 100 ng/mL. In these systems, the tracer was added simultaneously with one of the aforementioned first antibodies (AS#706: 1/80,000 and AS#Pool: 1/50,000). The following day, the double antibody precipitation system was added and a further 30 min incubation was carried out before separation of bound and free PAG.

The minimum detection limit (MDL), calculated as the mean concentration minus twice standard deviation (mean – 2 SD) of 20 duplicates of the zero (B_0_) standard [44] were, respectively, 0.4 ng/mL and 0.3 ng/mL for RIA-706 and RIA sr-Pool. Intra-assay CVs of RIA-srPool were 5.8% and 9.5%, respectively. Interassay CVs of RIA-706 and srPool were 5.0% and 3.0%, respectively.

### 2.5. Pregnancy Diagnosis

A cut-off value of 1.0 ng/mL was used for concentrations of progesterone or PAG (RIA-706, RIA-srPool) in order to distinguish between pregnant and nonpregnant females [12].

### 2.6. Statistical Analysis

Pregnancy-associated glycoprotein and progesterone concentrations were expressed as least square means ± standard error of the mean (±SEM).

The data obtained were analyzed using the GLM procedure of SAS (2013) [45]. A mixed model with repeated measures considering sheep breed (Sarda or Lacaune), RIA method (POV or 706), sampling time (day of mating and 18, 30, 45, 60, 90 and 120 days after mating; during the postpartum period starting within 12 h from lambing and then on days 7, 14, 21 and 28 of lactation), type of delivery (single or multiple) and sex of fetuses as fixed factors, and all possible interactions between main factors was used. The ewe was considered as a random factor.

Due to the small number of twin pregnancies, the effect of fetus gender were estimated for single pregnancies only.

Two interactions (sheep breed × RIA method and sheep breed × RIA method × sampling time) were removed from the model because the results of ANOVA were not statistically significant.

Differences between the least square means were evaluated by Tukey test (*p* < 0.05). Tendencies were discussed when *p* > 0.05 but ≤ 0.10.

The chi-square test was used to assess the agreement between the two RIA methods in early pregnancy (at 18 and 30 days after mating).

Finally, the elimination rate constant was calculated from the slope of the line during the postpartum period by linear regression analysis of the semilogarithmic plot of PAG concentrations versus time, while the half-life was obtained as (ln 2 /elimination rate constant).

## 3. Results

From a total of 35 synchronized Sarda and 35 Lacaune ewes, 26 and 20 animals became pregnant respectively as shown by the RIA analysis and the lambing rate.

Nine Sarda and 15 Lacaune sheep were diagnosed to be nonpregnant and were considered as negative controls. They gave PAG concentrations below the cut-off value.

Among the Sarda ewes, 20 had single pregnancies while six carried twins. The average length of gestation was 149.5 days (149.1 and 148.1 for ewes carrying one and two fetuses, respectively). There were 14 male fetuses.

In Lacaune ewes, 14 had single pregnancies while six carried twins. The average length of gestation was 146.0. There were 15 male fetuses.

### 3.1. Progesterone Concentrations

Progesterone concentrations were detectable in 26/26 (100%) and 20/20 (100%) of pregnant Sarda and Lacaune ewes, respectively. The mean progesterone level of nonpregnant Sarda and Lacaune ewes on Day 18 were 0.1 ± 0.1 ng/mL for both breeds. In pregnant females, progesterone levels ranged from 5.26 ± 0.7 ng/mL on Day 18 to 6.05 ± 0.5 ng/mL on Day 60 for Sarda ewes, and from 10.9 ± 1 ng/mL on Day 18 to 12.0 ± 0.7 ng/mL on Day 60 for Lacaune ewes.

### 3.2. Profiles of RIA-706 and RIA-srPool during Pregnancy and the Postpartum Period in Sarda and Lacaune Ewes

Similar PAG profiles were observed for both breeds. PAG plasma concentrations detected with RIA-706 were lower than the PAG levels detected with RIA-srPool (overall means: 21.82 ± 0.83 vs 30.49 ng/mL ± 0.83, respectively). However, a significant (*p* < 0.001) interaction between the RIA method and sampling time was detected and differences between the two methods were only recorded on days 45, 60, 75 and 90 during pregnancy, while no differences were observed during the postpartum period (Table 1).

Regarding the effect of breed, higher plasma PAG values were observed for Lacaune ewes than for Sarda ewes (overall means: 27.41 ± 0.90 vs 24.90 ± 0.86 ng/mL, respectively). When plotting the values of the interaction between sheep breed and sampling time together (*p* < 0.001), significant differences were only observed on days 75 and 90 (Figure 1) for both RIA systems.

No differences were observed during the postpartum period. For both RIA systems, at 28 days postpartum recorded values fell below 1 ng/mL (cut-off ≥ 1 ng/mL).

On day 18 of pregnancy the PAG concentrations in Sarda ewes were ≥ 1 ng/mL (cut-off value) in 25/26 (96.1%) and in 22/26 (80.8%) with RIA-706 and RIA-srPool, respectively. As described in Table 2, it was possible to detect PAG in all pregnant animals as early as Day 30.

Mean PAG-706 and PAG-srPool progressively increased from the day of conception (0.08 ± 0.02 and 0.06 ± 0.02 ng/mL, respectively) until day 60 (43.77 ± 5.64 and 73.62 ± 5.64, respectively) then decreased until stabilizing on day 90 (16.25 ± 3.64 and 27.46 ± 3.64, respectively), after which they increased and peaked on the day of delivery (56.77 ± 8.50 and 73.07 ± 8.50 ng/mL, respectively).

In this breed, both PAG-706 and PAG-srPool progressively decreased from parturition reaching values below 1 ng/mL at 28 days after lambing (0.39 ± 0.13 and 0.44 ± 0.13 ng/mL, respectively).

We used scatter plots of ln PAG-706 and ln PAG-srPool concentrations versus postpartum days to calculate the kinetic parameters. The elimination rate constants were 0.12 day^−1^ and 0.13 day^−1^ while the half-lives were 5.8 days and 5.3 days for PAG-706 and PAG-srPool, respectively.

On day 18 of pregnancy, the concentrations of PAG in Lacaune ewes were ≥ 1 ng/mL (cut-off value) in 18/20 (90%) and in 15/20 (75%) measured with RIA-706 and RIA-srPool, respectively. As described in Table 2, it was possible to detect PAG as early as day 30 in all pregnant animals.

Mean PAG-706 and PAG-srPool progressively increased from the day of conception (0.11 ± 0.03 and 0.06 ± 0.03 ng/mL, respectively) to day 75 (50.83 ± 6.65 and 85.51 ± 6.65 ng/mL, respectively), then decreased until day 120 (37.14 ± 3.34 and 35.33 ± 3.34 ng/mL, respectively), after which they increased and peaked on the day of delivery (57.86 ± 9.69 and 58.95 ± 9.69, respectively).

Both PAG-706 and PAG-srPool progressively decreased following parturition, but during the postpartum period at 28 days after lambing, PAG concentrations once again exceeded 1 ng/mL (1.45 ± 0.15 and ± 1.04 ± 0.15 ng/mL, respectively).

We used scatter plots of ln PAG-706 and ln PAG-srPool concentrations versus postpartum days to calculate the kinetic parameters. The elimination rate constants were 0.10 day^−1^ and 0.11 day^−1^ while the half-lives were 6.9 days and 6.3 days for PAG-706 and PAG-srPool, respectively.

### 3.3. Effects of Single Versus Multiple Pregnancies and Gender on PAG Concentrations

At delivery, in Sarda ewes multiple pregnancies were characterized by higher levels of PAG than single pregnancies (Figure 2 and Figure 3) disregarding RIA system (significant interaction breed × type of delivery × sampling time).

The PAG levels in plasma tended (*p* = 0.064) to be affected by the gender of the fetus (24.37 ± 0.78 vs 22.26 ± 0.84 for males and females, respectively). There were no significant interactions between gender of the fetus, sheep breed and RIA method.

## 4. Discussion

To our knowledge, this is the first study to be carried out on PAG plasma concentration profiles during gestation and the postpartum period in Sarda and Lacaune ewes, using two different radioimmunoassays, during the whole period of gestation and postpartum.

The PAG profile was similar in both breeds and in both RIA systems used. During the gestation period, PAG concentrations increased up to 60 days and then decreased until 120 days. Thereafter there was a significant rise, which peaked at parturition. This is a similar trend to that described by Gajewski et al. [46] for Berrichon ewes, by Ledezma-Torres et al. [14] for various sheep breeds and by Ranilla et al. [11,24] for Churra sheep, yet this trend was not observed for Merinos ewes whose PAG levels initially increased and then dropped to baseline concentrations around mid-pregnancy. Patterns of plasma ovine PAG concentrations differed from those reported for bovine [30], goats [47,48] and buffalo [20,49]. These differences could be explained by the ability of the antisera to distinguish between various epitopes [20].

In the first period of gestation, our data showed that both RIA (RIA-706 and RIA-srPool) are capable of diagnosing pregnancy as early as 18 days of gestation. Similarly, Ayad et al. (2007) [50] reported that it is easier to distinguish between pregnant and nonpregnant cows if a mixture of different antisera is used.

In Lacaune ewes, PAG concentrations measured by using RIA-706 were higher than those measured by RIA-srPool from parturition until the last day of observation (28 days postpartum). In Sarda ewes concentrations decreased rapidly reaching minimal levels (<1 ng/mL) at day 28 when using both RIA-706 and RIA-srPool. A similar decrease in PAG concentrations during the first month postpartum was reported by Ranilla et al. [11,24] for ewes, by Sousa et al. [41] and Gonzalez et al. [51] for goats, and for wild ruminants by Ropstad et al. [52] and Osborn et al. [53]. In these species, PAG concentration drops below 1 ng/mL at day 30 postpartum. In cows, PAG concentrations slowly decreased following parturition and were detected as late as 100 days postpartum [30]. The rapid decrease in PAG concentration during the postpartum period is essential when using PAG as an appropriate marker of pregnancy immediately after postpartum. Unlike in cows, the rapid PAG disappearance in ewes does not require the use of a cut-off limit in postpartum animals as a means for detecting a new pregnancy.

The half-life proved to be longer in Sarda and Lacaune sheep (6.3 to 6.9) than in other ovine breeds as reported by Haugejorden et al., 2006 [54] (4.5 days), or in buffalo females (5.8 days) [21,49]. However, it was shorter than in bovine species (from 7.0 to 8.8 days) [12,54,55,56] in goats (7.5 days) and Zebu (9.2–10.1 days) [57]. These differences could be due to the presence of N-linked carbohydrate and sialic acid chains on their structure [58].

This study demonstrated that mean PAG concentrations were higher in multiple pregnancies than in single pregnancies, at delivery in Sarda ewes. The higher concentrations in multiple delivery with respect to single delivery are possibly due to the higher member of attachment points, and thus secretory activity, of twin placentas [24]. Regarding the effect of the litter size on PAG concentration, our results showed for the Sarda sheep an evident peak of concentration at delivery, compared to Lacaune sheep, whether the RIA-706 or RIA-srPool was used. This behavior differs from that found in other breeds of sheep [3]. The structure of the placenta could be at the origin of this difference. In sheep before birth, a decrease in the number of binucleated cells is observed [59], and probably in the Sardinian sheep it could undergo a further increase in the prebirth period similar to what happens in cattle [42] and in Churra sheep [11]. However, a different profile during pregnancy was also found in the goat [60] and in the cow [61], depending on the breed.

From our data it was impossible to predict litter size. These results are in agreement with those of various authors [3,12,13,14,16,24,28] who were unable to predict litter size from PAG concentrations.

Our data show that PAG plasma levels tended to be affected by the gender of the fetus. This difference may be due to the weight of the placenta that differs according to gender [62,63]. Our results are in agreement with those reported by Guilbault et al. [64] and Zoli et al. [30] for bovine species, and by Ranilla et al. [11] for Churra ewes but not for Merinos ewes, and are in contrast to those reported by Vandale et al. [13] for Suffolk and Texel breeds, for which no significant differences were observed between ewes carrying fetuses of different gender. Moreover, our results suggest that the breed and the gender of the fetus could influence ovine PAG production.

## 5. Conclusions

In conclusion, this study showed, for the first time, the plasmatic profile of PAG in Sarda e Lacaune ewes during gestation and the postpartum period using two different RIA-systems, thus enhancing our knowledge regarding PAG concentrations in domestic ruminants. Moreover, it showed that for both breeds, these RIA systems are capable of distinguishing pregnant from nonpregnant ewes starting from day 18 of gestation. Furthermore, the rapid disappearance of PAG concentration following lambing did not require the use of a cut-off limit in postpartum animals as a means for detecting a new pregnancy.

## Figures and Tables

**Figure 1 animals-10-01502-f001:**
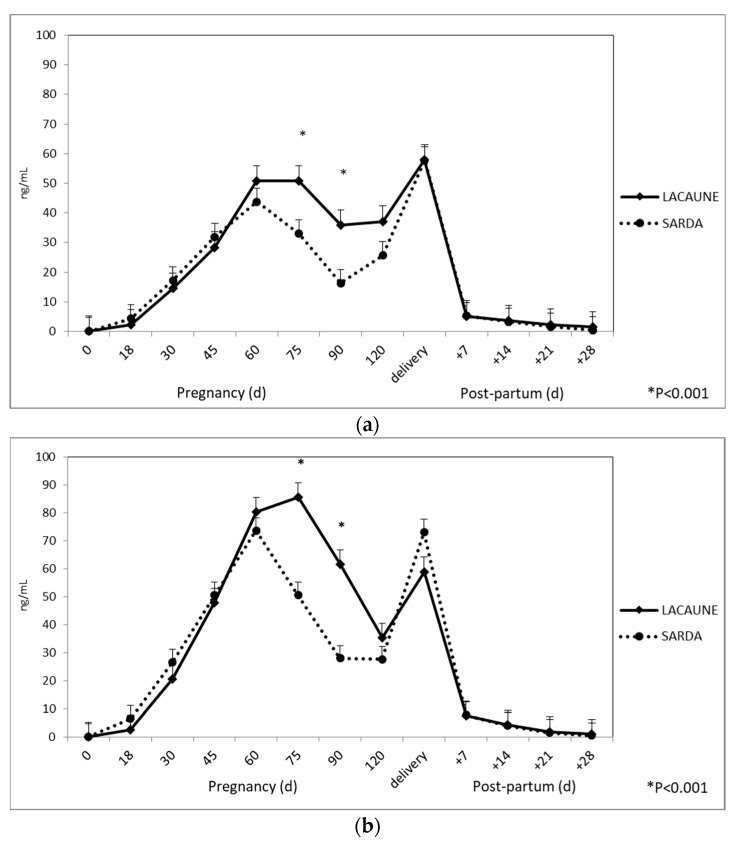
Plasma PAG profile in Lacaune and Sarda ewes during pregnancy and postpartum when using (**a**) RIA-706 and (**b**) RIA-srPool.

**Figure 2 animals-10-01502-f002:**
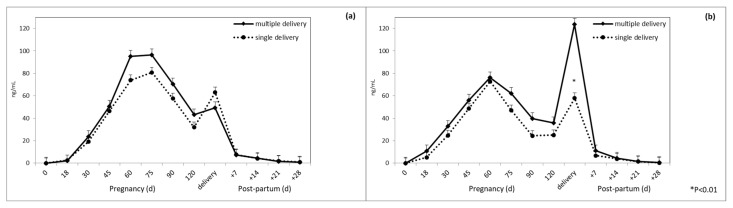
Plasma PAG profile in multiple and single pregnancy at delivery in Sarda ewes when using (**a**) RIA-706 and (**b**) RIA-srPool.

**Figure 3 animals-10-01502-f003:**
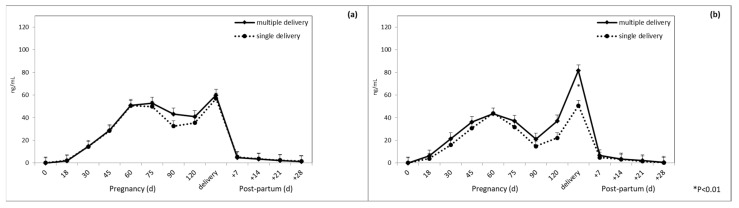
Plasma PAG profile in multiple and single pregnancy at delivery in Lacaune ewes when using (**a**) RIA-706 and (**b**) RIA-srPool.

**Table 1 animals-10-01502-t001:** Pregnancy-associated glycoprotein (PAG) values obtained before and after delivery according to radioimmune assay (RIA) methods.

Days	RIA-706	RIA-srPool	*p*-Value
Mating	0.148	0.059	0.23
18	3.735	4.775	0.69
30	16.907	24.034	0.59
45	31.298	50.189	<0.001
60	48.709	78.444	<0.001
75	44.836	69.821	<0.001
90	29.183	46.975	<0.001
120	33.852	34.171	0.33
Delivery	62.753	71.873	0.78
7	5.573	8.035	0.55
14	3.576	4.275	0.08
21	2.057	1.778	0.98
28	0.998	0.842	0.09

**Table 2 animals-10-01502-t002:** PAG detection (cut-off ≥ 1 ng/mL) in Sarda and Lacaune ewes according to RIA method (RIA-706 vs. RIA-srPool) at day 18 and day 30 of pregnancy.

Day of Pregnancy		Sarda		Lacaune	
		RIA-706	RIA-srPool	RIA-706	RIA-srPool
Day 18	Nonpregnant	1	4	2	5
	Pregnant	25	22	18	15
Day 30	Nonpregnant	0	0	0	0
	Pregnant	26	26	20	20

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
