# Peer review of "Plasmatic Profile of Pregnancy-Associated Glycoprotein (PAG) during Gestation and Postpartum in Sarda and Lacaune Sheep Determined with Two Radioimmunoassay Systems"

_animals, 2020, doi:10.3390/ani10091502_

Round 1

Reviewer 1 Report

Manuscript ID: animals-898993

Title: Plasmatic profile of pregnancy-associated glycoproteins (PAG) during gestation and post-partum in Sarda and Lacaune sheep determined with two radioimmunassay systems

By De Carolis et al.

General comments

The topic is interesting and the manuscript well presented. In opinion of this reviewer, the acceptance of the manuscript can be greatly supported. Just one point: please improve Figures

Minor points

L26 - please spell for the first time in the text PAG.

L44 – what day.

L114 – spell PMSG.

L205 – match please with the references section. (Ref 50?)

Reference 18 – please add the title

Author Response

Responses to Reviewer #1: Review Animals -898993

We thank Reviewer #1 for the appreciation of our paper. All the suggestions have been take into account to improve the manuscript.

General comments

The topic is interesting and the manuscript well presented. In opinion of this reviewer, the acceptance of the manuscript can be greatly supported. Just one point: please improve Figures

WE HAVE IMPROVE FIGURES, AS SUGGEST

Minor points

L26 - please spell for the first time in the text PAG.

            DONE; line number 27

L44 – what day.

            DONE; line number 45

L114 – spell PMSG.

            DONE; line number 125

L205 – match please with the references section. (Ref 50?)

            DONE; line number 209

Reference 18 – please add the title

            DONE; line number 428

Reviewer 2 Report

Well written manuscript.

Minor corrections as follows

  1. Line 109: mention the pelleted concentrate used along with contact details for the manufacturer;
  2. Line 136: mention the test kit used to extract progesterone along with the contact details for the manufacturer;
  3. Line 295 replace reference [14] with [50];

Author Response

Responses to Reviewer #2 Review Animals-898993

We thank Reviewer #2 for the appreciation of our paper. All the suggestions have been take into account to improve the manuscript.

Line 109: mention the pelleted concentrate used along with contact details for the manufacturer;

REVISED AS REQUEST; line number 119, 120, 121.

Line 136: mention the test kit used to extract progesterone along with the contact details for the manufacturer.

We did not use an extraction kit but a homemade method that has always been used in our RIA laboratory For greater clarity I have added other specifications of the method, which in any case is amply described in reference 38

Line number 139

Line 295 replace reference [14] with [50]

DONE; line number 304

Reviewer 3 Report

Dear author,

I have the following 2 comments: 

  1. In Metarials and Methods  2.1. Animals and samples it would be better if it can be splited in 2 parts with the 2.1 Animals and 2.2. Experimental design and not to include everything under the title you have.

2. In Discussion at Page 8, Line 295 you have to correct the number of the reference with the wright one to be  Ayad et al 2007, (50) instead of (14)

Author Response

Responses to Reviewer #3: Review Animals -898993

We thank Reviewer #3 for the appreciation of our paper. All the suggestions have been take into account to improve the manuscript.

Dear author,

I have the following 2 comments: 

In Metarials and Methods  2.1. Animals and samples it would be better if it can be splited in 2 parts with the 2.1 Animals and 2.2. Experimental design and not to include everything under the title you have.

REVISED AS REQUESTED. Line number 97, 113 and 131

In Discussion at Page 8, Line 295 you have to correct the number of the reference with the wright one to be  Ayad et al 2007, (50) instead of (14)

DONE; line number 304